# Teaching and Learning IoT Cybersecurity and Vulnerability Assessment with Shodan through Practical Use Cases

**DOI:** 10.3390/s20113048

**Published:** 2020-05-27

**Authors:** Tiago M. Fernández-Caramés, Paula Fraga-Lamas

**Affiliations:** 1Department of Computer Engineering, Faculty of Computer Science, Universidade da Coruña, 15071 A Coruña, Spain; 2Centro de investigación CITIC, Universidade da Coruña, 15071 A Coruña, Spain

**Keywords:** IoT, cybersecurity, Shodan, teaching methodology, use case based learning, security audit, vulnerabilities, cyber-attacks, vulnerability assessment

## Abstract

Shodan is a search engine for exploring the Internet and thus finding connected devices. Its main use is to provide a tool for cybersecurity researchers and developers to detect vulnerable Internet-connected devices without scanning them directly. Due to its features, Shodan can be used for performing cybersecurity audits on Internet of Things (IoT) systems and devices used in applications that require to be connected to the Internet. The tool allows for detecting IoT device vulnerabilities that are related to two common cybersecurity problems in IoT: the implementation of weak security mechanisms and the lack of a proper security configuration. To tackle these issues, this article describes how Shodan can be used to perform audits and thus detect potential IoT-device vulnerabilities. For such a purpose, a use case-based methodology is proposed to teach students and users to carry out such audits and then make more secure the detected exploitable IoT devices. Moreover, this work details how to automate IoT-device vulnerability assessments through Shodan scripts. Thus, this article provides an introductory practical guide to IoT cybersecurity assessment and exploitation with Shodan.

## 1. Introduction

The Internet of Things (IoT) is a paradigm that involves the connection to the Internet of daily objects, giving remote users and other devices the possibility of monitoring and interacting with them. According to some reports, 75 billion IoT devices will be deployed by 2025 [1] for multiple areas like smart appliances [2], smart agriculture [3], smart healthcare [4,5], or smart cites [6] (a summary of the most relevant IoT application areas is shown in Figure 1). Part of such areas are considered as critical, so their security is key to avoid potential damage.

Cybersecurity is a necessary requirement that has to be addressed during the design, implementation and deployment of IoT devices [7,8]. One of the most challenging problems of current IoT devices is that many of them are battery dependent and can be considered as resource-constrained in terms of computational power and memory, which prevents them from implementing certain security features that are common in traditional computers. For instance, public-key cryptography is essential for providing high security for web browsing [9], email exchanges [10], or for storing medical data [11], but the implementation of cryptosystems like Rivest–Shamir–Adleman (RSA) [12] or Elliptic Curve Cryptography (ECC) [13] may not be possible or inefficient for resource-constrained IoT devices. Moreover, such constrained devices may include bugs in their firmware, which in many cases is not possible or easy to update periodically with code patches.

Weak credential security and the lack of basic authentication measures are also common in IoT devices. For instance, such weaknesses were exploited by Mirai, which created a botnet that obtained the administrative credentials of other IoT devices through brute force. Mirai-infected devices, like webcams, Digital Video Recorders (DVRs), or routers, carried out in September 2016 one of the largest Distributed Denial of Service (DDoS) attacks in history, with hundreds of thousands of devices performing simultaneous requests [14]. In many cases, the mentioned weaknesses are related to the fact that, often, product development does not consider security until the final development stages, as an additional layer, instead of considering it as a design requirement.

Although there are a number of recent results of research projects that deal with IoT cybersecurity [15,16], it is almost neglected in many university degrees that are related to the development of IoT products (e.g., electrical engineering, computer science, and computer engineering), so graduated students do not receive in most cases a dense training on IoT security. Moreover, such a lack is also amplified by the difficulty of evaluating a broad range of real IoT devices, which would provide hands-on experience to the students.

To tackle the aforementioned lack, this article includes the following contributions:A practical use case-based teaching methodology is proposed. Such a methodology is based on Shodan [17], an online tool that accelerates significantly the IoT device reconnaissance stage, which is usually the most time and resource consuming stage on a cybersecurity assessment.This article also provides an introduction to the basics on IoT cybersecurity for future developers, which can harness Shodan Application Programming Interfaces (APIs) to build tools to automate IoT device vulnerability assessments.A theoretical and empirical approach to IoT security is provided to help educators to replicate the teaching results obtained by the authors, which have successfully put them in practice in seminars and master courses since 2018. For such a purpose, multiple practical use cases are provided together with useful guidelines to prevent Shodan-based attacks.

The rest of this article is structured as follows. Section 2 analyzes the most recent and relevant work on cybersecurity and IoT security teaching. Section 3 details the proposed teaching methodology. Section 4 details the basics on IoT cybersecurity, including the most common security concerns and the most popular IoT devices and architectures. In addition, Section 4 indicates the main IoT security attacks and describes the typical IoT audit/attack methodology. Section 5 details the basics on Shodan, and Section 6 suggests multiple use cases to put into practice the proposed teaching methodology. Finally, Section 7 is devoted to the conclusions.

## 2. Related Work

### 2.1. Cybersecurity Teaching and Learning

Despite the increasing importance of cybersecurity, it is currently not taught extensively in many universities around the world. Some universities have incorporated cybersecurity topics in their study programs [18], but it is still difficult to determine which core competencies should be imparted and then find experts to teach them [19].

Although most cybersecurity teaching still follows the traditional approach based on lectures and labs, some universities have taken them to the cloud and thus imparted virtual cybersecurity lectures on cloud-based platforms. For instance, the authors of [20] describe their experience when teaching a cybersecurity course across two campuses via a virtual classroom. The authors use the Amazon Web Services (AWS) cloud and remark as its main advantage that students perform the exercises in a contained and secure environment without having to deal with cumbersome tasks to set up and configure cybersecurity tools. A similar approach is detailed in [21], where the concept of Cybersecurity Lab as a Service (CLaaS) is proposed to provide cybersecurity experiments to students that can be anywhere and that only need an Internet connection and a device like a laptop, tablet, or a smartphone to carry out the required tasks of the course.

Commercial software and hardware can be used for recreating real-world scenarios for cybersecurity labs, but some researchers find them limited in different aspects and thus created their own frameworks. For instance, in [22], a cybersecurity framework is proposed to develop hands-on experiments rapidly, making use of two incentive models to engage the participants: a model to encourage engineers to contribute with data and experiments and a model to encourage universities to use the contributed data/experiments for education. A different approach is followed in [23], where researchers from Northumbria University (United Kingdom) propose a low-cost and flexible platform that is used as honeypot and that can be integrated with general purpose networks. Similarly, in [24], a modular testbed for teaching cybersecurity in a simulated industrial environment is presented. By using a flipped classroom methodology, students learn about threats associated with the industrial control system domain, develop an educational game, and exercise their soft skills during multiple public presentations.

Regarding IoT cybersecurity teaching, there are not many well documented success cases in the literature. An example is detailed in [25,26], where a course in secure design is described. Such a course is aimed at teaching students how to make user-centered cybersecure products that communicate threats in a better way and that emphasize key decisions to the user. The course consists of classroom instruction, hands-on labs, and prototyping tasks where the students build a conceptual model of a popular IoT smart home product.

Practical experimentation seems to be essential in IoT cybersecurity learning, as it allows the students to retain the knowledge longer than when only traditional lectures are given [27]. For instance, practical experiments carried out with the hardware platform Proxmark3 are key when teaching [28] and evaluating [29] Radio Frequency Identification (RFID) cybersecurity.

Apart from hands-on assignments, other approaches to cybersecurity training include serious games [30]. Examples range from cybersecurity competitions with penetration testing practices [31], capture the flag games [32,33,34], online learning platforms [35], red versus blue teams [36], or build-it/break-it/fix-it competitions [37]. In this regard, Hendrix et al. [38] investigate whether serious games can be effective cybersecurity training tools. Although their results are generally positive, the authors remark that the evaluation sample size was small and selected. Moreover, the studied games were designed for a very short-term interaction (to be finished in one session), and those papers that included an evaluation only considered immediate short-term impact. Therefore, although the authors considered the positive early indications, the question of whether serious games are effective at training was difficult to answer conclusively. As a result, they concluded that games could represent specific case studies and facilitate case-based learning approaches.

Finally, it is worth mentioning that the vast majority of the IoT cybersecurity literature is aimed at training/teaching university students, but it is also important to consider younger students, who are progressively being taught to code from a younger age. This is why the authors of [39] analyzed potential security and privacy issues that may arise when teaching children how to program the BBC micro:bit platform, which can be used by kids to build their own IoT devices. Other authors focused on promoting training all age groups and on further engaging female students [40]. In such a paper, the authors emphasize the role of problem solving using the scientific method and experiential learning activities.

In contrast to some of the previously mentioned IoT security initiatives, this article proposes to make use of a tool that can be used remotely by any student with just a device able to run a web browser and an Internet connection. Therefore, there is no need for expensive hardware or cloud infrastructure (in the imparted courses, students with smartphones were able to perform most of the methodological steps as if they were using more powerful computers). In addition, although the proposed methodology was specifically conceived for university students, it can be easily adapted to high school teaching. However, it must be pointed out that the practical use cases described later in Section 6 allow for detecting many real-world exposed IoT devices, including some related to industrial or critical scenarios, which may lead to access voluntarily or involuntarily IoT devices and networks that belong to third parties. Therefore, every student/researcher/teacher should check and follow the respective law of his/her country and, of course, not cause any trouble or damage to the involved IoT systems.

### 2.2. Shodan for IoT Cybersecurity

There are different web-based search engines for generic vulnerability scanning like Zmap [41] or Censys [42], and other online tools like Thingful [43] that are used for gathering data from connected IoT devices, but Shodan is currently the best suited for learning IoT cybersecurity due to the ease of use of its web and API interfaces.

In the last years, several researchers made use of Shodan to evaluate the security of different IoT devices. For instance, in [44], the authors used Shodan to detect devices like routers, firewalls, or web cameras that made use of default credentials or simple passwords. Similarly, in [45], Shodan was used together with other tools like Masscan and Nmap to detect vulnerable DSL routers, printers and IoT devices affected by the Heartbleed bug. In the case of [46], webcams and connected smart cameras were the ones analyzed: the researchers found thousands of them poorly configured or with no security. Other researchers corroborated such results and concluded that webcams are in general barely protected and can be used for cyberattacks [47]. Even more concerning are the results of the work detailed in [48], where numerous vulnerable medical devices were detected using Shodan.

It is also worth mentioning the survey in [49], which emphasizes the need for hardening IoT device security at the view of the ease of use of Shodan and the existence of tools like ShoVAT [50], which automate vulnerability identification. Nessus [51] can also be used for vulnerability identification together with Shodan [52]. Such an assessment can also be carried out through scripts, like the authors of [53] did back in 2014 to detect thousands of exposed webcams, printers, and even traffic control systems. Finally, it must be noted that IoT security analyses can be restricted to certain physical locations or organizations. For instance, in [54], the authors scanned IoT vulnerabilities in Jordan, finding numerous open webcams, industrial control systems and automated tank gauges.

## 3. Teaching Methodology

This article proposes to structure the learning/teaching process into four main parts:Introduction to the main IoT cybersecurity concepts. In this first part, the basics on IoT topics like IoT communications architectures, common IoT devices, and attacks to IoT systems are addressed.Introduction to the vulnerability assessment tool. This second part deals with the basics on the use of Shodan.Practical use case-based analysis. A set of use cases is given to the students in order to apply to them the proposed analysis methodology. At this point no knowledge of computer programming is required, only a web browser with access to Shodan.IoT audit/attack automation. In this final part the students learn how to develop scripts to automate the cybersecurity assessments that in the previous part they performed manually through the Shodan web interface.

The first three of the previous four parts can be carried out by most students that have a minimum knowledge of computers and IoT. Nonetheless, the methodology obtains better results with computer science and electrical engineering students, who usually have a good previous knowledge on how IoT devices and architectures work.

The previously mentioned structured content is typically imparted in an intensive six-week course. Each week, one and a half hours are dedicated to theoretical lectures and another one and a half hours to practical labs. In addition, the students carry out a guided final project on the security of a specific device or field. Although the students choose freely the theme of the project, they are guided by the course instructor to make the most out of the learning experience.

It is important to note that the proposed teaching structure is not lineal throughout the course: most of the theoretical concepts are given during the first three weeks, whereas the last three weeks are essentially focused on the labs and on the final project. Thus, the last three weeks are taught in a flipped classroom format [55], where students are given additional content (e.g., links to IoT security presentations from conferences like DEF CON [56], BlackHat [57], or CCC [58]) that are later discussed during the face-to-face time.

At the end of the course, the students deliver three reports and the corresponding software for the labs and for the final project. The grades are given as follows: 40% of the grade is related to an exam on the theory, 30% is for the lab reports, and 30% is for the final project.

The following syllabus was proposed during the imparted courses:Essential IoT cybersecurity Part I (theory, week 1).
Introduction to IoT.Traditional IoT architectures.Advanced IoT architectures.Shodan basics (lab 1, week 1).
Introduction to Shodan.How Shodan works internally.Shodan basic use.A first search with Shodan.Essential IoT cybersecurity Part II (theory, week 2).
Popular IoT devices.Main components of an IoT device.Main IoT-device security problems.Practical IoT security analysis with Shodan (lab 2, week 2).
Analysis methodology.Practical use cases.–Webcams.–Home automation systems.–Home devices.Essential IoT cybersecurity Part III (theory, week 3).
Common IoT-device vulnerabilities and attacks.Shodan query automation (lab 3, week 3).Final project (weeks 4-6).

It is important to note that teachers should emphasize throughout the lectures the importance of the legal dimension and possible consequences of putting Shodan and similar cybersecurity tools to practice. The next sections of this article provide details on the main topics of the previous syllabus.

## 4. Essential IoT Cybersecurity

### 4.1. Main Concerns on IoT Security

As it was previously mentioned in the Introduction of this article, the security of many IoT devices is conditioned by their computational simplicity and their dependence on batteries. The former prevents developers from using security mechanisms that require relevant amounts of computing power or memory, while the latter deters them from implementing complex cryptosystems that can drain the battery fast. There are high-security energy-efficient mechanisms [59], but their implementation is not very common in commercial IoT devices.

Static memory is also a common problem in IoT resource-constrained devices, as software bugs and misbehaviors can be discovered after the deployment stage and thus require to patch the device firmware. Unfortunately, many IoT devices (e.g., sensors and actuators) have not been designed to be updated, like the ones based on Application-Specific Integrated Circuits (ASICs) or whose firmware is stored on a Read-Only Memory (ROM). Other devices are difficult to update for most users, such as the IoT devices that require to disassemble the device and plug a specific hardware programmer in. Nonetheless, it must be mentioned that some IoT devices (usually the most computationally powerful, like smart TVs) can be updated via Over-the-Air (OTA) updates, which allow for receiving periodic firmware patches, dynamic configuration settings, or encryption keys from an IoT provider or a user.

Although most IoT users are essentially concerned by end-device security, IoT networks are composed by other devices like gateways or remote clouds that are also vulnerable to attacks. As an example, Figure 2 shows, on the right, the main components of a traditional IoT cloud-based architecture, which is currently the most popular among commercial IoT deployments. Such an architecture consists of three layers. The layer at the bottom is the IoT-node layer, which is composed by IoT devices that collect data from their embedded sensors and that receives remote commands from the cloud. IoT nodes connect to the cloud through the gateway layer, which includes local gateways (e.g., wireless or wired access points) and gateways deployed by Internet-Service Provider (ISPs) to reach the Internet. Finally, at the top of the architecture is the cloud, which stores, processes, and provides access to the collected data and allows for sending commands to the IoT devices.

### 4.2. Traditional and Advanced IoT Architectures

Although cloud-based architectures are currently the most popular, they are related to certain security problems that can be prevented by using other advanced architectures. For instance, one of the problems of cloud-based architectures is that they concentrate most of the complex processing and storage on the cloud. This means that the cloud becomes a point-of-failure and, if it has a fault (e.g., due to a cyberattack, to periodic maintenance, or to a power outage), then the whole IoT system stops working properly. Moreover, when a lot of devices perform requests simultaneously, the cloud becomes a bottleneck that slows down the operation of the IoT network due to the excessive workload.

To tackle the previously mentioned issues, decentralized architectures based on edge computing are useful. Figure 2 shows, on the left, the main components of an edge computing based architecture, where three main layers can be distinguished: the IoT node layer, the cloud, and the edge computing layer. The IoT node layer and the cloud operate in a similar way to a traditional cloud-based architecture. The key layer is the edge layer, which provides edge computing services through fog computing gateways and/or cloudlets [60,61,62]. Fog computing gateways are often devices like Single-Board Computers (SBCs) that provide fast responses and some processing power to the IoT devices in order to reduce latency and the amount of network traffic that is forwarded to the cloud. Cloudlets have similar objectives, but they are usually high-end computers that perform computing intensive tasks locally. It is also important to note that edge computing nodes can communicate with each other, thus being able to collaborate among them to carry out specific tasks.

There are also other alternative architectures for deploying IoT systems, like the ones based on mist computing [63] or on blockchain [64], which are currently still being studied by industry and academia even for future post-quantum scenarios [65].

### 4.3. Popular IoT Devices and Cyberattacks

There are many traditional devices that have been enhanced by enabling new features by adding an Internet connection. This is case of TV sets, set-top boxes, home automation systems, intelligent light bulbs, or smart power outlets. Most of them make use of a cloud-based architecture that centralizes request processing in a remote cloud. In this way, if, for instance, a user wants to switch on an smart power outlet through a smartphone app, the user request is first sent to the remote cloud and then the cloud forwards it to the power outlet. This switch-on process is described step by step in Figure 3, where it can be observed that a number of potential problems can arise when user-to-cloud and power outlet-to-cloud communications security is either weak or neglected. Some examples are:An evil twin attack can be performed to create a fake local gateway that is able to route IoT device communications to another remote server.DoS or DDoS attacks can be performed on the cloud, thus preventing users from sending commands or receiving information from the IoT devices. Similar results may be achieved by carrying out such Dos/DDoS attacks on the communications gateways, which are usually less powerful and less prepared for supporting cyberattacks.Weakly encrypted or plain-text communications can be intercepted through sniffers or Man-in-The-Middle (MiTM) attacks, which can gather data on the user or on certain IoT device activities.Insecure IoT systems can also be affected by MiTM attacks that are able to modify commands or IoT device responses so as to change the expected behavior of the system.

The impact of the previously mentioned cyberattacks is not only related to traditional homes, but it is amplified due to the broad application fields where IoT is involved, like the deployments related to healthcare [66], smart cities [67], smart infrastructure [68], smart campuses [69], intelligent transport systems [70], or defense and public safety.

In addition, it is important to note that IoT devices like the smart power outlet included in Figure 3 are composed by three different components: hardware, software, and connectivity. Each of such components can be subject to specific attacks and vulnerabilities:Hardware attacks. This kind of attacks is related to vulnerabilities that affect certain hardware parts embedded into an IoT device. Examples of such attacks are:
–Physical attacks.–Battery/power removal.–Reverse engineering of the hardware.–Denial of Service (DoS) attacks to drain batteries.Software attacks. These vulnerabilities are related to software bugs or to certain misbehavior that lead to security problems. For instance, some software attacks of this type are:
–Software reverse engineering.–Software vulnerabilities that have or have not been properly patched.–Malicious software injection.–Weak cryptographic implementations.Connectivity attacks. As connectivity is key for implementing the IoT paradigm, IoT devices are vulnerable to traditional attacks aimed at intercepting the exchanged data or at triggering certain behaviors by impersonating an authorized third party. Thus, some of the most relevant connectivity attacks are:
–DoS attacks.–Jamming and radio interference.–IoT node impersonation and Sybil attacks.–Man-in-the-Middle attacks.–Network protocol attacks.

### 4.4. IoT Audit/Attack Methodology

Figure 4 illustrates the main steps of the proposed IoT audit/attack methodology, which essentially consists of four phases:Reconnaissance. In this phase the auditor/attacker gathers information on the IoT target. The collected data may come from multiple sources (e.g., manufacturers, IoT providers, and hardware datasheets) and includes the traditional port scanning process in order to determine which services are available.Audit/Attack plan. The auditor/attacker designs the steps involved in the devised audit/attack strategy and selects the most appropriate tools to implement the plan. In many cases it is necessary to develop specific tools to later exploit certain IoT device vulnerabilities.Access to the IoT system. The previously selected tools are used to access the IoT system. Such tools exploit hardware, software, or connectivity vulnerabilities.Execution. After accessing the system, an attacker/auditor will put in practice the previously planned strategy to take control of one or more IoT devices. It is common to make use of certain software mechanisms to maintain the access to the IoT system for future intrusions (e.g., by opening a backdoor).

Among the previously mentioned phases, the first one (reconnaissance) is usually tedious and requires to dedicate a significant amount of time and resources. However, as it is detailed in the next section, thanks to Shodan, this stage can be noticeably shortened.

## 5. Shodan Basics

### 5.1. Aims and Inner Working

Shodan is actually a search engine that scans the Internet IP by IP looking for available services. Such services are detected by parsing banners, which are essentially text that allow for identifying login interfaces or certain service characteristics. An example of a banner is the typical Secure Shell (SSH) login interface, which may provide details on the software of the SSH server or on the computer where it is executed. Shodan indexes banner information and then allows for consulting it through a web interface (shown in Figure 5) and programming APIs.

Underneath, Shodan makes use of crawlers that gather data continuously. There is a crawler network that operates in different countries to prevent IP geo-blocking. Each crawler execute a really simple script that carries out the following steps [71]:

A random IPv4 is generated.A random port is selected among the ones supported by Shodan, which are usually related to essential services.The crawler tries to connect to the select IP and port, and if a connection is established, it collects the banner.Go back to step 1.

### 5.2. Basic Use and Web Interface

Shodan can be used as any web search engine, but its use and results differ depending on the user role: there are non-registered users, free registered users, and paid registered users. Each type of user can perform a different number of requests per month, scan a limited number of IPs, and monitor a network with a different maximum of IPs. Differences also exist on the use of certain features, like the application of certain filters or the provided support. Researchers, educators, and students that register with an academic email address can receive a free upgrade (which needs to be requested by email) that enables accessing enough functionality to teach/learn how to use Shodan, but that is usually limited to not to use the accounts to develop commercial applications.

To illustrate the features of Shodan with an example, it can be for instance searched for “*openwrt*”. OpenWrt [72] is actually an operating system based on Linux for embedded devices that can be executed in IoT networks by devices like routers, SBCs, Network Attached Storage (NAS) servers, WiFi extenders, or webcams. The previous Shodan search will lead to a screen like the one shown in Figure 6, which indicates the most relevant sections of the result list page.

When the user clicks on Shodan Maps, the web interface shows a map like the one shown in Figure 7, where the estimated location of the detected OpenWrt devices is depicted. Figure 8 shows the extended information for one of the results obtained in the search. In this screen, on the left, for some devices, detected vulnerabilities are shown. The collected raw data can be accessed by clicking on “View Raw Data”.

Among the multiple features included by Shodan, filters are one of the most useful when looking for specific IoT devices. The following are some of the most relevant:country: it specifies the country of the detected devices through an ISO 3166-1 alfa-2 code. For instance, if the previous Shodan search was meant to be limited to the United States, the following query text should be indicated: *“openwrt country:US”*.city: it indicates the city of the devices to be located. For instance: *“openwrt city:Barcelona”*.geo: it allow for filtering the results depending on their geographical coordinates. If, for instance, the previous results were aimed at obtaining the OpenWrt devices that are located next to Paris city center, the Shodan search would be: *“openwrt geo:48.860151,2.336200”*. Moreover, this filter can received a third parameter that indicates the maximum radius of the search. For example, the previous search can be modified to obtain the devices that are in a circle of one kilometer around coordinates 48.860151, 2.336200: *“openwrt geo:48.860151,2.336200,1”*.net: it filters the results according to an IP range indicated in Classless Inter-Domain Routing (CIDR) notation. An example would be: *“openwrt net:37.13.0.0/16”*.port: it allows for filtering the results depending on the detected open ports. For instance, the following Shodan query would return the OpenWrt devices whose port 21 (FTP) is open: *“openwrt port:21”*.org: it filters the results according to the organization they belong to. As an example, the following query would indicate the OpenWrt devices that are managed by Amazon: *“openwrt org:amazon”*.

More filters and their parameters can be found in [73].

## 6. Practical IoT Security Use Case Analysis with Shodan

### 6.1. Use Case Analysis Methodology

#### 6.1.1. Teacher Perspective

From the teaching point of view, the following methodology would be recommended:As a first step, the teacher will give the students a list of Shodan searches (like some of the given in Section 6.2).Basic analysis. The students analyze the results obtained by each query and determine which IoT device they are looking for and what it is used for. This process usually involves multiple Google searches to look for vendor information like device manuals/datasheets.Vulnerability assessment. The students study the vulnerabilities detected by Shodan, they look for default credentials and for other potential cybersecurity problems.

As an example, the previously detailed methodology can be applied to a popular webcam software for Microsoft Windows:First, the teacher would give the students the following Shodan query without giving further details on the IoT device: *webcamxp*.Next, the students would introduce the query in Shodan and would find out that several thousands of results (more than 5000 as of writing) are shown, most of which are related to a webcam software. As Shodan currently returns a relevant number of honeypots, the students would have to make use of filters to retrieve real webcams. For instance, a refined Shodan search would be: *product:“webcamXP httpd”*.After applying the appropriate filters, it is not difficult to find open webcams like the one shown in Figure 9 on the right. It is also straightforward to find further information on the software by looking for *webcamxp manual* through a web search engine.Finally, the students will look for security vulnerabilities of the IoT device. In this specific case, the vast majority of the detected webcams neither make use of passwords or implement any kind of access restrictions to control the webcam. The cybersecurity of the hosts that make use of each webcam can be further analyzed with the help of Shodan (e.g., open ports or services), but such a traditional analysis is in general out of the scope of a course focused on IoT cybersecurity.

#### 6.1.2. IoT Researcher Perspective

Determine the target IoT device.Build the Shodan search. This first step requires to determine the most appropriate query and its filters in order to obtain the desired list of target IoT devices.Look for additional information on the target IoT device. This process may involve looking for information provided by the manufacturer or for the default credentials indicated in the user manual.Vulnerability assessment. In this step it is necessary to analyze the vulnerabilities found by Shodan, the security data provided by the manufacturer or already published Common Vulnerability and Exposure (CVE) reports.

The WebcamXP example given in the previous subsection for the teacher perspective can be used to illustrate how the proposed methodology would be applied by an IoT researcher:First, the researcher would set as an objective to find vulnerable webcams that make use of WebcamXP software.Next, the researcher will design a first Shodan query (for instance, *webcamxp*) to retrieve the maximum possible amount of IoT devices. Once a webcam is successfully detected (like the one shown in Figure 9 on the right), the Shodan search can be easily refined to avoid collecting data from honeypots and from other devices that include the word *webcamxp* in their banner. For such a purpose, the researcher can analyze the raw information collected by Shodan and select certain fields and values that are highly likely to remain constant for most of the targeted IoT devices. For instance, filtering out by product (Shodan query: *product:“webcamXP httpd”*) or by certain fields of the HTTP header (Shodan query: *“Pragma: no-cache Server: webcamXP”*) can be useful.At this point, the researcher may be interested in finding more information on the possibilities that the webcam software can bring to a remote auditor/attacker. For such a purpose, further information on the webcam software is available on the WebcamXP user manual, which can be easily found through a web search engine.Although most of the WebcamXP webcams found through Shodan are completely open, the researcher may be interested in exploring further security vulnerabilities of the detected IoT devices. In such a case, CVE repositories like CVE Details allow for searching for WebcamXP vulnerabilities [74], showing three CVE reports: CVE-2008-5862, CVE-2005-1190, and CVE-2005-1189. Shodan academic users can make use of the mentioned CVE IDs and Shodan’s vulnerability filter to obtain vulnerable devices directly (Shodan query: *vuln:CVE-2008-5862*).

### 6.2. Practical Use Cases

#### 6.2.1. Webcams and Video Surveillance Systems

Webcams and video surveillance systems probably provide the most common examples on how users lack of knowledge on IoT device security affects privacy and security around the globe: it is currently very easy to find unprotected webcams and video surveillance systems that use their default credentials. Examples of Shodan queries to find this kind of systems are:Linksys WVC80N Wireless Internet Camera (Shodan query: *WVC80N*). This is a webcam for home monitoring that is more than 10 years old, but that still is serving in homes and industrial installations. The problem is that many users either use the default credentials (admin/admin) or do not use authentication at all, which causes a privacy problem (an example of screenshot obtained from an open WVC80N webcam is shown in Figure 9 on the left).ExacqVision (Shodan query: *“server: wfe”*). This is a video surveillance system that allows for watching and managing multiple webcams through a web interface. The problem is that a significant number of users do not configure authentication or make use of weak/default credentials.AXIS webcams (Shodan query: *“port:80 has_screenshot:true”*). As of writing, more than 3000 of these webcams can be found through Shodan, many of them requiring no credentials to watch them.AVTECH IP webcams (Shodan query: *linux upnp avtech*). More than 180,000 AVTECH devices can be currently found by Shodan with the previous query, although many of them require credentials to access the video stream. Although the latest firmware versions ask for a verification code, there is a significant number of webcams that make use of the default credentials (admin/admin).

#### 6.2.2. Home Automation Systems

The presence of home automation systems whose security is neglected is also significant. The following are some examples of Shodan queries that will retrieve open or weakly protected home automation system:JUNG KNX (Shodan query: *Jung KNX*). This is a home automation system whose smart control panel can be accessed remotely with no need for credentials (an example of such a smart panel is shown in Figure 10 on the left).Jeedom (Shodan query: *Jeedom*). It is a French open-source home automation system that usually provides a web interface and, in many cases, an open Message-Queue Telemetry Transport (MQTT) broker.Somfy alarm system (Shodan query: *title:“Centrale" Pragma:"no-cache, no-store”*). The previous search allows for locating thousands of Somfy alarm systems, which provide a web interface for remote user authentication.Insteon home automation system (Shodan query: *title:“powered by insteon”*). Most of the Insteon installations located through the previous Shodan search require no authentication, so remote users can interact directly with them (a example of an already hacked system is shown in the screenshot in Figure 10 on the right).Creston control hub (Shodan query: *Crestron PYNG-HUB*). The web panel of this hub is used by hundreds of users to monitor and control their home automation devices.

#### 6.2.3. Home Devices

Like in the case of home automation systems, many home IoT devices are weakly secured or not secured at all. Some examples of interesting Shodan queries are:iKettle (Shodan query: *ikettle*). It is a smart appliance to boil water remotely.WebIOPi (Shodan query: *webiopi*). It is a framework for creating and deploying IoT applications with Raspberry Pi. Many installations are not password protected (an screenshot from one of such installations that monitors environmental temperature is shown in Figure 11 on the left).Open Virtual Network Computing (VNC) systems (Shodan query: *has_screenshot:true product:VNC “authentication disabled”*). The previous query allows for detecting VNC systems whose authentication has been disabled.MQTT brokers (Shodan query: *“MQTT Connection Code: 0” set –alarm*). Although MQTT is very popular among IoT developers, its security, in many cases, is neglected. Thus, the previous Shodan query finds a significant number of open MQTT brokers.Yamaha AV receiver (Shodan query: *“HTTP/1.1 406 Not Acceptable” “Server: AV_Receiver”*). Many Yamaha Internet-enabled AV receivers, which provide a remote web interface, have disabled their authentication (a screenshot of one of them is shown in Figure 11 on the right).

### 6.3. Automating Attacks

#### 6.3.1. Shodan APIs

Shodan web interface provides a fast way to perform general evaluations on IoT devices or on very specific use cases. However, to automate use case analysis, the APIs provided by Shodan are more appropriate.

Currently, Shodan provides APIs for Python, Ruby, PHP, C#, Go, Haskell, Java, Node.js, Perl, PowerShell, and Rust. Two specific APIs are defined: a Representational State Transfer (REST) API and a streaming API. The REST API is aimed at interacting with Shodan through GET, POST, DELETE, and PUT requests. The streaming API is able to exchange data that is embedded into JavaScript Object Notation (JSON) files. Shodan also provides an additional REST API for exploits [75], which normalizes exploit information after collecting it from multiple vulnerability data sources.

#### 6.3.2. Teaching Shodan Scripting

In order to learn how to automate the manual steps described in Section 6.1, the following tasks can be performed by students/learners:Install the code development environment. This usually requires importing Shodan search library.Perform an initial Shodan query through the code to find a specific version of an IoT device.Modify the code in (2) to print the IP and country of every obtained result.Modify the code in (3) to print, for each detected IoT device that has vulnerabilities, the number of detected exploits according to Shodan exploit REST API.

For instance, the following steps would be needed to perform the previous four tasks when using Python:First, it is necessary to install Python and then install the Shodan module with the command *“pip install shodan”*.An example of the script required for carrying out step 2 is shown in Listing 1 (between lines 1 and 15). Note that, in order to execute the script, it is necessary to indicate the Shodan API key of the developer. In the example, the indicated query can be changed to adapt to the user needs.Listing 1 also shows the part of the script to perform step 3 (between lines 1 and 25). It is worth noting that a 1 second delay is needed, as Shodan may limit the number of requests to one per second.Step 4 can be implemented in Python with the code below line 26 of Listing 1, which makes use of the exploit REST API.

Listing 1: Example of Python script to automate Shodan queries.

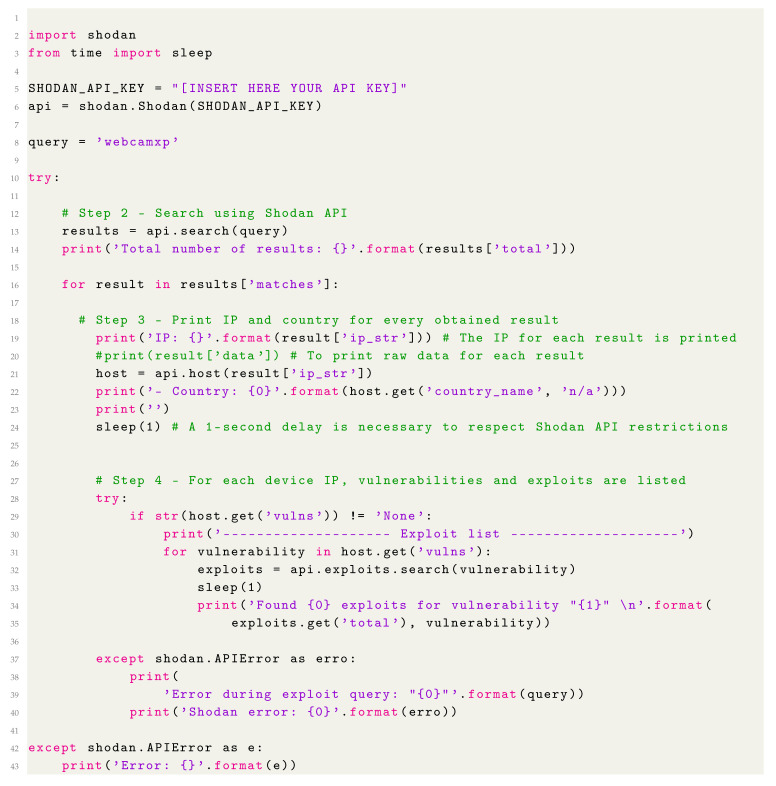



### 6.4. Practical Teaching Results

During the last years, the previously described methodology was taught at the University of A Coruña to students of the cybersecurity master program. Each student received three individual Shodan queries and had to first apply the methodology described in Section 6.1.1, and then learned to automate such queries through scripts, following the steps indicated in Section 6.3.2.

As a reference, the following paragraphs summarize the results obtained by the students of the 2020 class where 16 students (two women and fourteen men) took part in the course. Most of them were recent graduates from computer science and electrical engineering programs with good coding skills and basic knowledge on cybersecurity, but almost no previous experience on IoT. They also had no previous practical experience with Shodan.

Sixteen reports were delivered, with an average of 33 pages per report.Different Shodan queries were performed to target 16 specific IoT devices.On such 16 IoT devices, 675 non-patched vulnerabilities were found related to already published CVEs.Roughly 320 IPs and their running services were analyzed making only use of the information provided by Shodan (no additional scanning tools were used).Of the 320 analyzed IoT devices, 87 of them required no credentials to access private data or to manage the device. Moreover, 21 of them made use of the default user or administrator credentials. These results indicate that roughly one out of three analyzed IoT devices could be easily accessed by a remote attacker.

As an example, Table 1 summarizes some of the most relevant results obtained by the students. The following are the main conclusions that can be withdrawn from such results:
Mootools-based webcams:
Shodan query: (*“webcam 7”* OR *“webcamXP”*) *http.component:“mootools”-401*Relevant results:
–All the 20 analyzed webcams required no credentials to view their content.–Seven of the webcams were used as surveillance cameras in industrial scenarios, while 4 of them were aimed at watching road traffic in specific areas. In addition, 5 of the cameras were used as home surveillance systems. The other 4 webcams were used for monitoring public spaces.–Of the 20 analyzed systems, four of them made use of services and software affected by 66 vulnerabilities that were already documented as CVEs.Insteon smart home controller:
Shodan query: *title:“powered by insteon”*Relevant results:
–Only 19 results were obtained. Most of the IPs were located in Taiwan and were deployed in homes.–Of the 19 IoT systems, 15 of them required no credentials to interact with the smart home system.Somfy alarm system:
Shodan query: *title:“Centrale” Pragma: “no-cache, no-store”*Relevant results:
–Several of the analyzed systems made use of the default credentials, so attackers could access the alarm system and enable or disable it at will.IoT Proliphix thermostats:
Shodan query: *title:“Status & Control”*Relevant results:
–A relevant number of the studied IoT systems either used the default user or administration credentials, so a remote attacker could easily watch and manipulate the thermostat.Tesla PowerPack system:
Shodan query: *http.title:“Tesla PowerPack System”*Relevant results:
–Some of the analyzed IoT systems could be accessed as administrator by making use of the default credentials. However, most of the systems found through Shodan were actually classified as honeypots.Cannon VB-M600 network camera system:
Shodan query: *title:“Network Camera VB-M600” “200 ok server: vb”*Relevant results:
–Of the 20 analyzed systems, nine of them could be accessed with no credentials, while four made of use of the default credentials.–The software used by these systems were affected by 359 vulnerabilities documented through already published CVEs. Such vulnerabilities were essentially related to the use of outdated versions of Linux and Apache Tomcat.Twonky media server:
Shodan query: *“product:TwonkyMedia UPnP” http.title:“Twonky Server”*Relevant results:
–All the devices found through the indicated Shodan query were completely open, so remote attackers can access the shared media content.

Given these results, as one of the students indicated in his report, “it can be concluded that Shodan is a really powerful cybersecurity tool that is able to expose IoT device misconfigurations and vulnerabilities in an easy and fast way; the possibility of using Shodan for automatic IoT vulnerability assessments emphasizes the importance of taking care of security during IoT device installation and configuration, and makes it necessary to patch their software periodically”.

Finally, it is worth mentioning that, during the course, there were no major problems respect to the use of Shodan. The only relevant issues arose in relation to the following two topics:API-based development. During the development of the scripts the students had problems when dealing with the Python wrapper API, as part of it was not properly documented.Critical infrastructure vulnerabilities. In case of finding vulnerabilities that affected critical infrastructures, the students were told to warn the instructor so that he/she could take the appropriate measures (e.g., to warn the company/entity through the university on the encountered problems). For instance, during the course, the mentioned procedure was used by a student that found VoIP communications system of a military company that used the default credentials.

### 6.5. Preventing Shodan-Based Attacks on IoT Devices: Best Practices

The previous sections show that Shodan is a really powerful tool for performing IoT cybersecurity audits and attacks. In the case of the former, an auditor can give the following recommendations to prevent the audited IoT devices from being attacked through Shodan:Check your IP or your organization IP range to determine whether your IoT devices are already indexed by Shodan. If they are indexed, verify their connectivity needs, trying to minimize the number of them that accept incoming connections.Minimize the number of open ports. In addition, make use of firewalls to prevent potential intrusions.Always try to use HTTPS instead of HTTP. This may be difficult to implement in certain resource-constrained IoT devices. In addition, please note that it is very complex to have an individual (no self-signed) certificate for each IoT device, so try to implement additional security layers.Whenever possible, try to use a Virtual Private Network (VPN).Whenever possible, modify your IoT device banners and the exposed ports to make the reconnaissance stage difficult for potential attackers. For instance, move the necessary ports to a range that is not scanned by Shodan crawlers.Block Shodan crawler IPs to prevent IoT devices from being indexed. A good list of such crawler IPs can be found in [76].In case the IoT device cannot be protected from being indexed by Shodan:
–Never use default or really common credentials (e.g., “admin”, “1234”).–Try to use long usernames and passwords to avoid brute-force attacks.–Update credentials periodically.–Keep IoT device firmware updated.

### 6.6. Additional Course Topics

This article described the content, structure, and methodology applied to a 6-week course that, due to time restrictions, is focused on detecting vulnerable IoT devices that are publicly exposed on the Internet. However, a complete IoT cybersecurity program should extend the proposed syllabus and address other relevant topics, like:Ethical hacking. Students should learn about the implications and differences among black hat, white hat, and gray hat hackers, which can make use of Shodan with different purposes.Legality. Cybersecurity researchers and students should be fully aware of the legal dimension and potential consequences of making use of Shodan and other security tools.Defense against IoT attacks. Although Section 6.5 enumerates different recommendations to protect IoT devices against Shodan-based attacks, IoT devices are exposed to many more attacks, like the ones indicated in Section 4.3. Therefore, it is necessary to teach students how to protect IoT devices from physical attacks, software/hardware reverse engineering, malicious firmware updates, or rogue wireless access points.Critical infrastructure cybersecurity. IoT devices can be deployed in environments whose infrastructure can be considered as strategical or critical due to the impact that cyberattacks can have on them. For instance, cyberattacks on certain industries (e.g., chemical plants and power plants) or infrastructure (e.g., bridges, dams, ports and railways) can have terrible consequences, so students need to be trained on the specific characteristics of such environments and on the most commonly used monitoring devices (e.g., Programmable Logic Controllers (PLCs) and Industrial Control Systems (ICSs)).Mobile device security. A mobile device, like a wearable, a smartphone, or a tablet, can be considered as a specific type of IoT device that provides users with certain communications services and monitoring capabilities (e.g., by making use of embedded sensors like accelerometers, gyroscopes, and GPS). For instance, unfortunately, Shodan can find thousands of open Android devices (Shodan query: *port:5555 debug*) that require no credentials for accessing the internal memory, for installing new applications, or for taking pictures with the embedded camera. Therefore, students should understand how the most popular mobile operating systems and devices work, and how they can be protected against cyberattacks.Platform security. Robot, cobot, Unmanned Aerial Vehicle (UAVs), or Augmented/Mixed/Virtual (AR/MR/VR) platforms can be considered as IoT platforms that make use of sensors, actuators, and communications subsystems that are expected to suffer from cybersecurity attacks. Students should understand how to keep information protected, defend against unauthorized use, tampering, or even physical damage.

## 7. Conclusions

IoT cybersecurity is a topic whose importance has been growing in the last years, but that has not been extensively covered in IT university programs. To ease IoT cybersecurity teaching, this article proposed a practical use case-based methodology that relies on Shodan, a search engine for exploring the Internet that is able to find connected IoT devices. Thus, students only need a web browser and Internet connectivity to carry out practical cybersecurity audits and analyses. Multiple practical examples have been given to discover IoT-enabled devices like webcams or home automation systems, which usually make use of default credentials and/or of weak authentication mechanisms. In addition, the article showed examples of scripts that allow for using Shodan to automate IoT-device vulnerability assessments. Thanks to the previous contributions, this article provided teachers and developers the basics for creating future Shodan-based IoT cybersecurity courses and vulnerability assessment software.

## Figures and Tables

**Figure 1 sensors-20-03048-f001:**
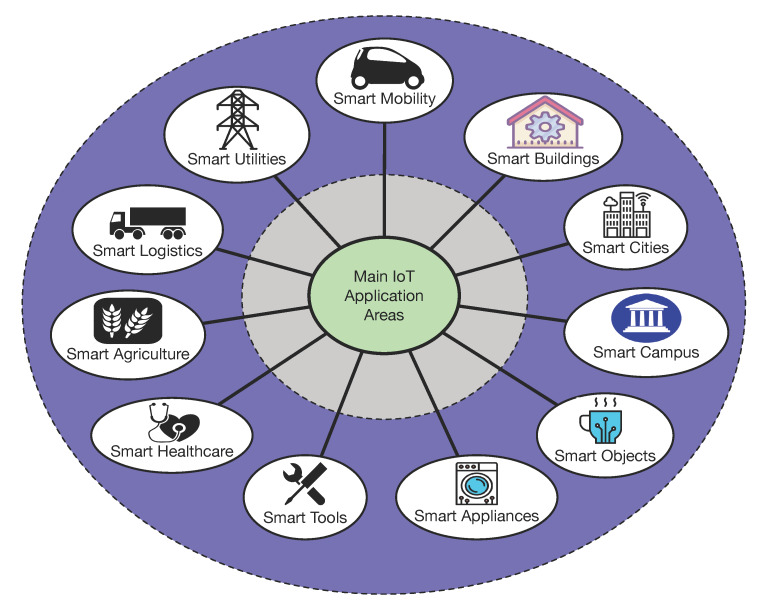
Main Internet of Things (IoT) application areas.

**Figure 2 sensors-20-03048-f002:**
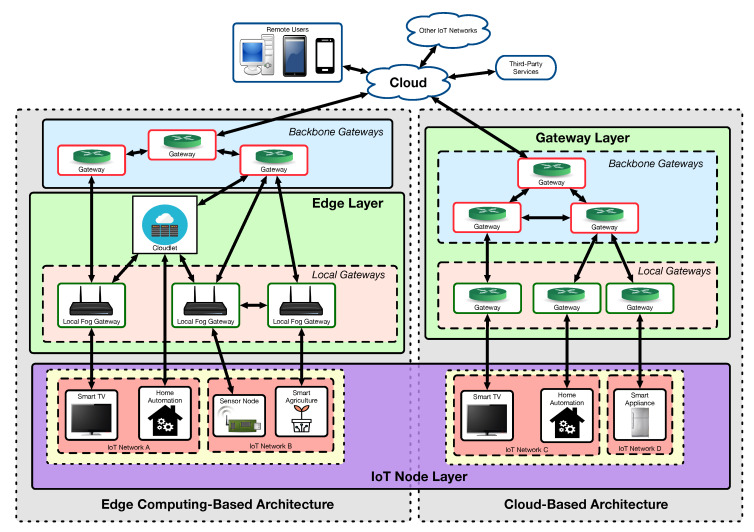
Components of cloud-based and edge computing-based IoT architectures.

**Figure 3 sensors-20-03048-f003:**
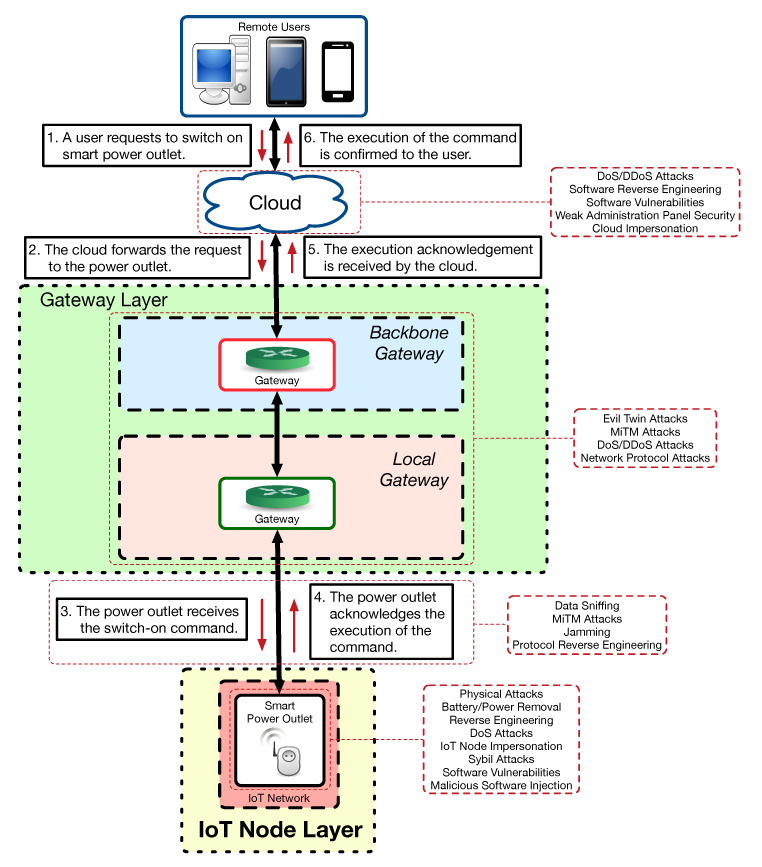
Switching on an IoT-enabled power outlet using a cloud-based architecture.

**Figure 4 sensors-20-03048-f004:**
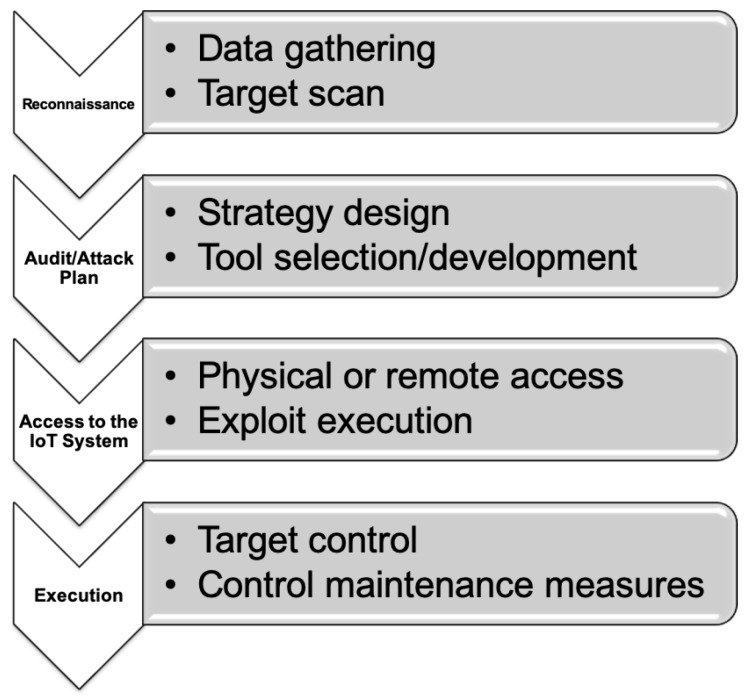
IoT device audit/attack methodology.

**Figure 5 sensors-20-03048-f005:**
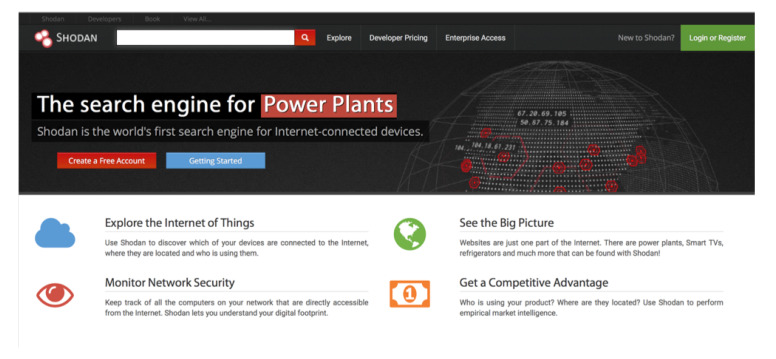
Shodan main web page.

**Figure 6 sensors-20-03048-f006:**
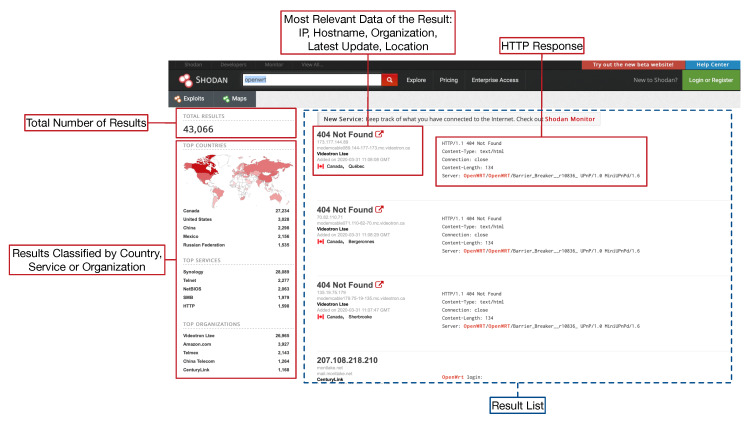
Example of Shodan result list page.

**Figure 7 sensors-20-03048-f007:**
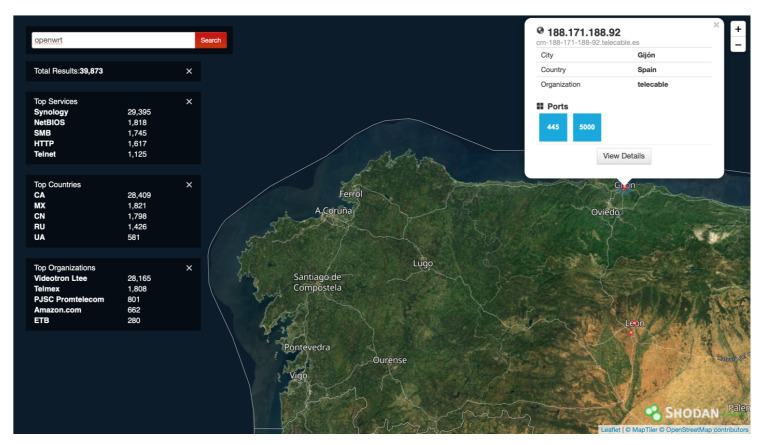
Shodan Maps interface.

**Figure 8 sensors-20-03048-f008:**
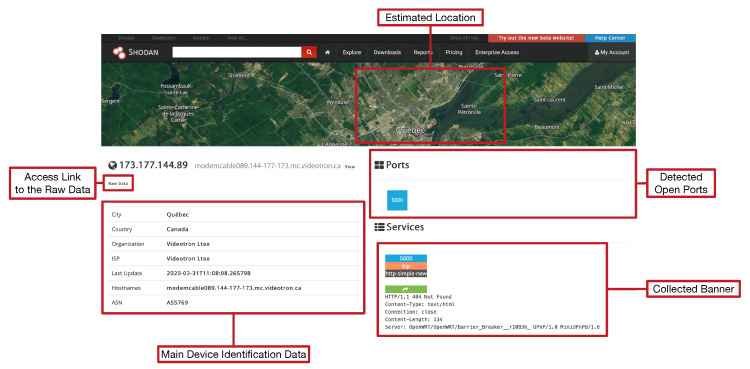
Shodan individual result data page.

**Figure 9 sensors-20-03048-f009:**
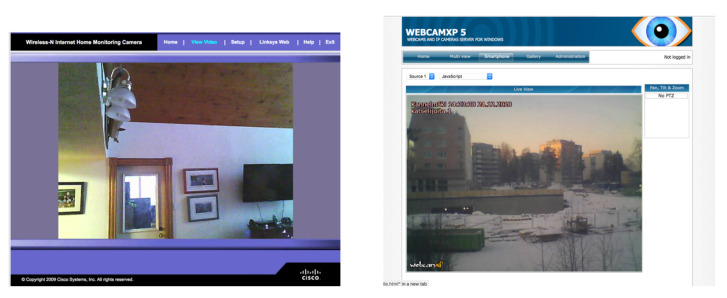
Screenshots of open WVC80N (**left**) and WebcamXP (**right**) webcams found with Shodan.

**Figure 10 sensors-20-03048-f010:**
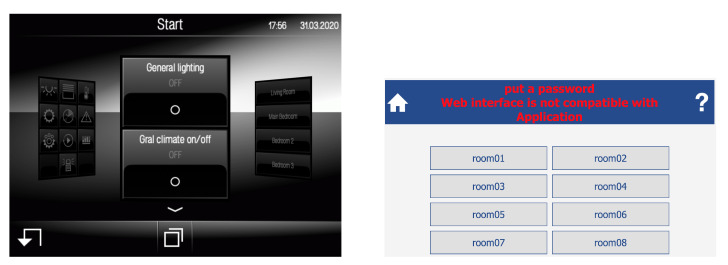
Screenshots of open Jung KNX (**left**) and Insteon (**right**) home automation systems.

**Figure 11 sensors-20-03048-f011:**
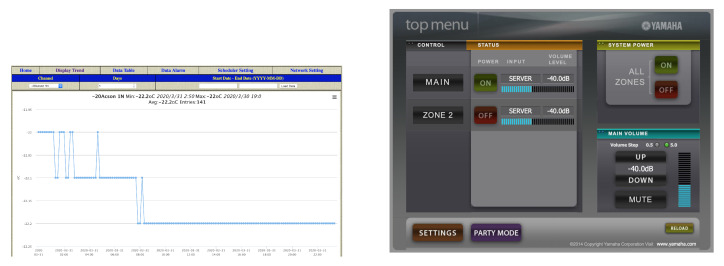
Screenshots of open WebIOPi (**left**) and Yamaha (**right**) installations.

**Table 1 sensors-20-03048-t001:** Summary of the most relevant results obtained by the students of class 2020.

IoT Device	Mootool-Based Webcams	Insteon Smart Home Controller	Somfy Alarm System	IoT Proliphix Thermostats	Cannon VB-M600 Network Cameras	Twonky Media Server
#Shodan Results	141	19	17,294	192	51	3846
#Analyzed Devices	20	19	20	20	20	20
#Devices without Authentication	20	15	-	-	9	20
#Devices with Default Credentials	-	-	2	3	4	-
#Devices Affected by CVEs	4	-	-	-	1	-
#Detected CVEs	66	-	-	-	359	-

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
