# Peer review of "Teaching and Learning IoT Cybersecurity and Vulnerability Assessment with Shodan through Practical Use Cases"

_sensors, 2020, doi:10.3390/s20113048_

Round 1
Reviewer 1 Report
The paper is well-written (apart from some minor spelling typos, see comments below) and is easy to follow. However, this reviewer has the following comments and feedback, which should be addressed before the paper can be considered for publication:
The authors should spell out clearly the main motivations and contributions of this work in the Introduction. The motivations related to teaching cyber security are slightly generic and the point made in section 4.5 about reconnaissance being the most time and resource consuming step in cyber-attacks on IoT devices should be highlighted in the Introduction.
In Fig. 4, instead of asking the reader to guess the potential problems in the user-to-cloud and cloud-to-IoT device chain, this should be elaborated on in the paper. The authors should use this example as an opportunity to illustrate the potential vulnerabilities and the security risks at each point in the chain. This section can be organised better by focussing on this scenario to motivate the central theme of the paper and Fig.2 and 3 can be combined into 1, showing the traditional/edge cloud architectures as the 2 possible options. Also, section 4.4 can be moved before Fig. 4 and then the figure can be annotated to show which types of attacks are applicable to the given smart home scenario.
Section 6.2: while the list of queries for searching for different IoT devices in different scenarios is quite comprehensive, the paper can be improved if all the steps of any one use case can be shown in detail, either through the student or researcher perspective.
In this regard, the results detailed in section 6.4 are quite informative for a reader of this paper.
While the results and discussion are informative, the authors should provide steps for future development of this work, listing how IoT researchers can use Shodan or similar tools to interrogate personal IoT deployments against specific cyber threats, such as WiFi access points against man-in-the-middle attacks.
Abbreviations should be defined on first use, e.g. SBC is first mentioned in section 4.2, but the full form only appears in section 5.2.
Too many self-citations, 18 out of a total of 80 – this is quite a high percentage. The authors should consider reducing the number of self-citations. For instance, for fog computing architectures, instead of referring to teaching-oriented papers, they can consider the following articles:
- C Perera, Y Qin, JC Estrella, S Reiff-Marganiec, Athanasios V Vasilakos, Fog computing for sustainable smart cities: A survey. ACM Computing Surveys (CSUR), 2017
- B Alturki, S Reiff-Marganiec, C Perera, S De, Exploring the Effectiveness of Service Decomposition in Fog Computing Architecture for the Internet of Things. IEEE Transactions on Sustainable Computing, 2019
Typos:
in Fig. 4, acknowledgement is spelled wrong
radio -> radius, section 5.2, line 358
surverillance -> surveillance (section 6.2.1, line 405)
wether -> whether (section 6.5, line 615)
In -> if (section 6.5, line 616)
Author Response
Dear Sir/Madam,
The authors would like to thank the reviewer for his/her valuable comments, which have certainly helped us to improve the manuscript. Please find attached our detailed responses to the comments. In order to ease the labor of the reviewers we have colored in red the differences with the previous version of the article.
Regards.

Reviewer 2 Report
The paper presents the teaching and learning of IoT cyber security and vulnerability assessment with Shodan. Overall, the paper is well written and documented.
As I personally have prior expertise with Shodan and cyber security teaching with Shodan, I appreciate the effort and I find the approach worth publishing. Nevertheless, there could be some additional chapters/topic added:
1. As this is a cybersecurity course, the students should also be taught about Ethical Hacking. Furthermore, it is important to mention, throughout the lectures, the legal dimension and possible consequences of putting these tools to practice.
2. more existing works and projects about securing IoT systems should also be mentioned. For example, the work "From Internet of Threats to Internet of Things: A Cyber Security Architecture for Smart Homes", and "GHOST - Safe-Guarding Home IoT Environments with Personalised Real-Time Risk Control" should be considered.
Some minor aspects:
- There are a few typos.
- Figures 2 and 3 should be placed side by side to observe the differences.
Author Response

(The authors gave the same response as above.)

Round 2
Reviewer 1 Report
The authors have revised the paper to address this reviewer's comments and the paper has been sufficiently improved. I believe this paper will be a good addition to the field of IoT/cybersecurity, both for students and teachers.